# Physical Visitor Access Control and Authentication Using Blockchain, Smart Contracts and Internet of Things

Frederick Stock [1], Yesem Kurt Peker [2], Alfredo J. Perez [3,*] and Jarel Hearst [2]

1  Department of Computer Science, University of Massachusetts Lowell, Lowell, MA 01854, USA
2  TSYS School of Computer Science, Columbus State University, Columbus, GA 31907, USA
3  Department of Computer Science, University of Nebraska at Omaha, Omaha, NE 68182, USA
*  Correspondence: alfredoperez@unomaha.edu

**Abstract:** In this work we explore the use of blockchain with Internet of Things (IoT) devices to provide visitor authentication and access control in a physical environment. We propose the use of a "bracelet" based on a low-cost NodeMCU IoT platform that broadcasts visitor location information and cannot be removed without alerting a management system. We present the design, implementation, and testing of our system. Our results show the feasibility of implementing a physical access control system based on blockchain technology, and performance improvements over a similar system proposed in the literature.

**Keywords:** blockchain; physical access control; access control; internet of things; physical security; wearables; IoT security





## 1. Introduction

Blockchain is a decentralized, digital ledger first introduced in 2008 as a transaction ledger for Bitcoin [1]. A blockchain consists of a series of "blocks" which can be thought of as pages in a traditional physical ledger. Each block is cryptographically linked such that a change to one block requires changing every following block. The design of blockchain makes it possible for systems to be tamper-resistant and protected from many outsider and insider threats, and it has spawned many applications in the realms of cryptocurrency [2], finance [3], healthcare [4], supply chain management [5], Internet of Things [6], crowd-sensing [7], digital art [8], among others. Blockchain's distributed nature makes it suitable for systems in need of resiliency and fault-tolerance requirements, and there are many applications that are currently being researched to take advantage of these properties. With these much-desired properties, Blockchain technology has become a subject of study in many application areas.

In this work we propose a prototype system for visitor physical access control based on blockchain, the Internet of Things, and smart contracts technology. Advantages that blockchain-based physical access control systems offer over centralized systems include avoidance of a single-point of failure, avoidance of malicious data modification (including modifications done by insiders), auditability, tamper-resistance, among others [9]. These advantages can make blockchain-based physical access control systems more reliable and trusted over centralized systems. While many systems have been proposed for physical access control throughout the years using centralized solutions, only recently blockchain-based systems have started to be researched for access control [10].

Our contributions in this work are as follows:

1.  Design of a prototype system for physical access control system using Blockchain, IoT devices, and smart contracts.
2.  Development of a prototype wearable wristband to assist in the authorization and tracking of visitors/users.

3.     Implementation and evaluation of the prototype system using the Ethereum platform.

The remainder of this paper is organized as follows. In Section 2 we review the literature on access control, including Blockchain-based physical access control. Section 3 describes our proposed model and implementation in detail. In Section 4 we present the results from our implementation of the model. Section 5 describes security analysis of our system. In Section 6 we provide a discussion on the overall system. Finally, in Section 7 we conclude our paper and provide possible ideas for future research.

## 2. Related Work

Access Control (AC) management systems have been studied in the context of centralized and decentralized systems. As centralized systems, early research dates back to the 60's and 70's with research done by Corbató [11], Graham and Denning [12], Lampson [13], and Saltzer [14] who researched access control for multiuser, time-shared systems. In the 90's with growth of the Internet systems, the Web, and commodity hardware and software, research in this area was further expanded with the development of Role-Based Access Control (RBAC) [15,16] and similar research [17,18]. In addition to centralized systems, most of the research conducted in AC systems before 2010 was conducted to support logical (data) AC, with implementations done to control Operating Systems (OS) and user resources (e.g., access to files, applications, and devices such as printers).

The dawn of the Internet of Things (IoT) has expanded AC research into cyberphysical systems using both centralized and decentralized systems [19–23]. IoT systems bring a new set of features that AC must deal with including a large amount of traffic/data generated by IoT devices, dynamic environments with new devices/users changing, a concern for privacy, and multiple vendors/standards that work together to meet design and application requirements [20,23]. The implementation of these AC systems for IoT can be done currently in as centralized systems in the cloud [24–28], or by using blockchain technology [10,29–34].

In the realm of blockchain systems for IoT access control, Novo [29] presents an architecture for IoT access control management based on blockchain and smart contracts. Their implementation makes use of blockchain nodes combined with management hubs to integrate resource-constrained IoT devices in a global access control network. In their work, smart contracts are used to define policies and operations in a private blockchain network. This prototype architecture and implementation makes use of managers who control IoT devices. No external users (different to the managers) are part of the prototype architecture. Zhang et al. [30] make use of a smart contract-based system for IoT AC to implement policy-based and dynamic access control policies. IoT devices are connected to gateway devices which then become peers in a blockchain network. Ding et al. [31] propose an Attribute-based Access Control (ABAC) blockchain for IoT AC. They argue that due to the large amount of devices and unknown identities, ABAC is more suitable for IoT device management. Liu et al. [32] examine the usage of blockchain to apply access control policies to the data collected by IoT devices. They propose a system with three types of smart contracts. The first type (device contracts) stores the URL to the data. Storing the URL instead of the full data helps with the rapidly growing blockchain storage size. The second type of smart contract (access contracts) determines whether a user is allowed to access the data on a device contract. Finally, policy contracts are used to modify and create access policies. They implement their model on Hyperledger Fabric. Their results show that their proposed method can "maintain high throughput in largescale request environment and reach consensus efficiently in a distributed system to ensure data consistency". Abdi et al. [34] designed a blockchain based IoT security and management system addressing the problems with traditional access control systems such as single point of failure, low scalability, and lack of privacy. Their proposed model is light- weight and reduces the computational load on the IoT devices and provides scalability. They propose a clustering approach with three main components: an Edge Blockchain Manager (EBCM) responsible for authenticating and authorizing constrained devices locally; an

Aggregated Edge Blockchain Manager (AEBCM) containing various EBCMs to control different clusters and manage access control policies, and a Cloud Consortium Blockchain Manager (CCBCM) that ensures that only authorized users access the resources. Smart contracts are used in the model to self-enforce decentralized access control policies. A proof of concept implementation of their proposed system on the permissioned Hyperledger Fabric shows that their solution is efficient and effective.

Li et al. [33] proposed an identity authentication system for constrained devices called Intelligent Electronic Devices (IEDs) found in the Chinese power system. They use a consortium blockchain to store credentials required for authentication. For authentication they propose hash chains which are one-time authentication passwords and require only one hash calculation in the authentication process without the need to deploy any third party for key management. This provides the much needed lightweight computations and efficiency in IoT systems. An automatic regeneration method of the hash chain is also proposed for the continuity of the identity authentication.

The studies briefly described above are mainly for AC for computer system resources (no for user authentication/physical access). For more information on blockchain-based AC for IoT devices, we recommend the surveys by Rouhani et al. [35], Abdi et al. [19], Qui et al. [20], Bagga et al. [36] and similar ones.

While there has been extensive work for blockchain-based AC for IoT devices, this has not been the case with blockchain-based physical access control, IoT devices, and smart contracts. We review recent work in this topic by Rouhani et al. [10], Mayle et al. [37], Chan et al. [38] and Bindra et al. [39].

Rouhani et al. [10] developed a system for physical access control based on RBAC, Hyperledger (a consortium-based blockchain), and smart contracts. Hyperledger is used in their system to provide storage for transactions (logs), as well as to define roles and permissions in the system. In their implementation, it is assumed that the authentication mechanism for physical access is provided by smart cards/smart card readers. Mayle et al. [37] propose a security system using MultiChain to build a private blockchain. Multi-Chain is built on the core Bitcoin, and by using a concept called smart filters in MultiChain they build a security system using microwave sensors and visual imagers (e.g., cameras). The implementation included five admin nodes and eight member nodes. The admin nodes have the authority to create and approve filters and add new nodes to the system and the member nodes represent the imagers, sensors, validators, and network admins in the system. Their results confirmed the resilience of the blockchain-based system against power failures, and it was also able to detect tampering of data and disregard the data associated with the tampering. Chan et al. [38] propose a system for visitor access control in a physical environment based on the public Ethereum blockchain platform in which a token is assumed to periodically scan a fingerprint of the visitor and get the visitor's location. These data are then submitted to a node on the blockchain.

Bindra et al. [39] propose an access control system for smart buildings using blockchain smart contracts to describe, grant, audit, and revoke fine-grained permissions for building occupants and visitors. Their system uses blockchain for physical access control involves four steps: (1) creating a unified Resource Description Framework (RDF) graph of a building by aligning the building's Build To Operate (BOT) and Brick (a metadata model for smart building) models; (2) identifying all possible paths between two locations using a graph traversal algorithm; (3) determining the cost of each path (4) granting, revoking, and verifying user permissions to access rooms and equipment therein using smart contracts. They implement their system using a private Ethereum blockchain which is then simulated in a building with five conference rooms.

From our literature review, we found that while there has been significant work on blockchain technology for IoT logical access control, not much work has been done in the study of decentralized, blockchain-supported physical access control systems thus far. Our work differentiates from the past works by making use of a prototype wearable device that serves a dual-purpose to authenticate and identify visitors while interacting

with a blockchain. While we use a private Ethereum blockchain and smart contracts in our system, we use a proof-of-authority algorithm, instead of a proof-of-work (or proof-of-stake) algorithm that increases the system's performance while keeping the system relatively lightweight and secure.

## 3. Materials and Methods

In this work we propose and implement a system that uses a private blockchain for controlling visitors' access to a physical site. In this section, we provide a description of the system, how the components of the system interact with each other, and details of our implementation of the system.

### 3.1. Proposed System

The proposed system consists of four main components: visitor devices, access points (APs), blockchain, and a management system as depicted in Figure 1. In the system the blockchain is assumed to be started by a trusted authority. The nodes of the blockchain are access points that are spread around the site. It is also assumed that two contracts, one for visitor registration and one for access policies are already deployed on the blockchain. The access policy in our implementation is based on levels. Levels are assumed to correspond to types of visitors. Regular visitor, privileged visitor, and other types as may be appropriate for the application. The levels are assumed to map to certain areas of the site that can be identified by ranges of longitudes and latitudes. In our implementation, we assume a hierarchical level structure in which higher level access allows access to areas that are in the lower level access.

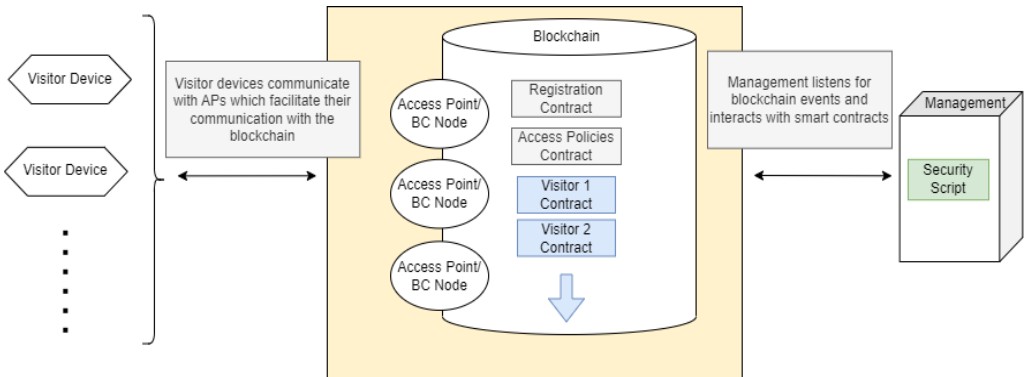

**Figure 1.** Overview of system architecture.

When a visitor arrives, their identity and access level are recorded in a special bracelet device which we call the visitor device. The visitor device, via the access points, communicates with the blockchain and registers the user via the registration contract on the blockchain. The registration contract issues a visitor contract on the blockchain that records the identification data as well as the access level of the visitor. The visitor device needs to be always worn by the visitor during their visit; it becomes the authentication token for the visitor as they tour around the site. The device is designed in such a way that its removal breaks the authentication and issues an alert to management indicating that the bracelet is no longer in place.

The visitor contract periodically receives requests to obtain updates about its corresponding visitor's location. If a visitor's location is not updated or is found to be in an area they are not authorized for, the smart contract emits an event which is received by the management system and appropriate action is taken by the security team on site. Figure 2. provides an overview of the interactions between the components of the system.

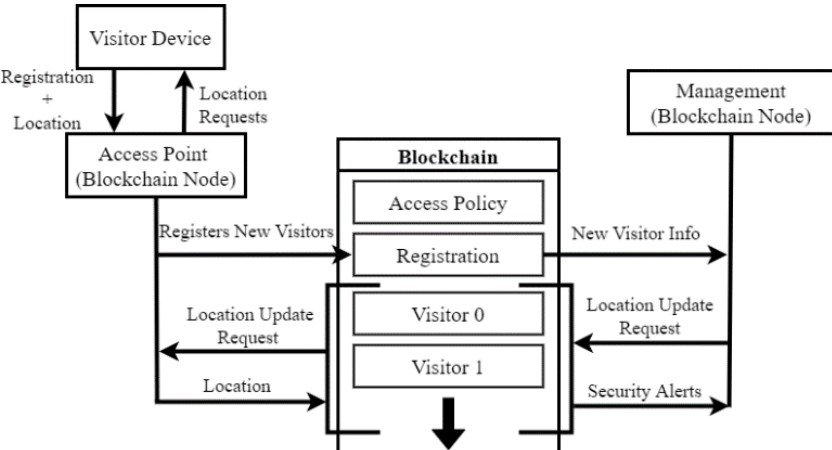

**Figure 2.** Overview of interactions between system components.

The main components of the system and their roles can be summarized as follows

- Visitor devices: Register visitors with the management system and report visitors' location.
- Access points: Request updates from visitor devices and connect with blockchain nodes.
- Blockchain: Serve as both the access control system and data storage (database).
- Management system: Serves as an administration interface for the system. It communicates with the blockchain nodes via smart contracts.

### 3.2. Implementation

#### 3.2.1. Visitor Devices

In our proposed system, a visitor device registers the visitor in the system, reports the visitor's location to the system, and assures that the visitor is indeed at the reported location with a high certainty. The registration of a visitor starts with the site authority setting up the visitor device and loading it with the information about the visitor including the access level for the visitor. The visitor device registers the visitor in the system by sending a transaction to the registration contract on the blockchain via an access point. The registration transaction creates a contract on the blockchain for the visitor associated with the visitor device sending the registration request. This "visitor contract" includes identifying information for the visitor, their access level, and their current location. After registering the visitor in the system along with their access data, the visitor device reports the location data to the system via the access points.

We implemented the visitor device as a wearable bracelet using a NodeMCU device with a basic circuit as seen in Figure 3. The circuit is simply a loop between a power and a ground pin on the board. A wire connects the A0 pin to this circuit, enabling the NodeMCU to quickly detect if this circuit is broken. A wire is looped around the visitor's wrist such that when the visitor removes the device from their wrist, the circuit between the power and ground pin will break. If the circuit breaks, the NodeMCU sends an alert to the management system via an access point.

The device determines and transmits its locations using NodeMCU's Wi-Fi capabilities and the Google Geolocation API. The device continuously computes the visitor's location at frequent intervals. When a visitor device receives a request to update its recorded location, it sends the most recently computed location to the access point, which is in turn logged in the system as long as it is authenticated as explained in the next paragraph.

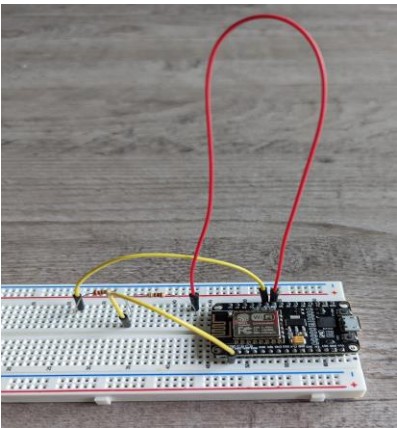

**Figure 3.** Prototype wearable visitor device based on a NodeMCU open IoT platform. The red wire must be worn around the visitor's wrist.

All transactions from a visitor's device are authenticated with a hashchain created and stored locally on the NodeMCU. A hashchain is the ordered list of hash values obtained by repeatedly hashing a random value. The authentication via hashchain is done by publishing the hash values one by one starting with the last one in the list and confirming that the hash of a newly published value is the same as the previously published hash value. In our implementation the last value of the hash chain stored on the device is submitted as part of the registration and stored on the visitor contract. Every message from the visitor device after that includes the next hash value from the tail of the hashchain and is submitted to the visitor contract in the updateLocation transaction (Figure 4) to authenticate the visitor device. If the hash of the new value sent in the transaction is different from the currently stored hash value, then the message is not coming from the visitor device and an alert is issued. Otherwise, the stored hash value is replaced by the new value. With this authentication, we can assume the only entity capable of updating a visitor's location is the corresponding visitor device assuming the hash function used is cryptographically secure. In our implementation we used SHA-256.

```
1  function updateLocation(bytes memory nextHash, uint64 newLat, uint64 newLong){
2      if(Sha256(nextHash) != currentHash){
3          emit securityAlert();
4      }
5      locations.push(newLat);
6      locations.push(newLong);
7      APContract.CheckAccess(newLat, newLong, accessLevel);
8      emit updatedLocation(tokenID, newLat, newLong);
9  }
10
11 function RequestNewLocation(){
12      emit provideCurrentLocation();
13 }
```

**Figure 4.** Listing for updateLocation and RequestNewLocation functions of VC.

### 3.2.2. Access Points

The visitor devices are assumed to be simple, cheap devices and hence do not have the capability to directly interact with the blockchain, necessitating an intermediate device. Visitor devices use access points as an intermediary to send information to the blockchain. In addition to providing a network access point for visitor devices, access points serve as nodes on the blockchain. In our implementation, we used Raspberry Pi devices as access points which are an affordable and practical option that can be deployed at various places on a site. The visitor devices and access points communicate using the MQTT protocol.

The MQTT broker can be run on one of these such access points or an entirely separate device depending on the requirements of the implementation.

The access points, in addition to communicating with the blockchain, are responsible for requesting updates from the visitor devices. Access points run a Python script that enables two-way communication between visitor devices and the blockchain. The script listens to the central MQTT broker and when a message is published from a visitor device, the access point passes the message onto the corresponding contract via a transaction. The script also listens for events from the blockchain and when an event is emitted which requires a visitor to respond, such as a request for a new location, the script notifies the appropriate visitor device by publishing a message through the MQTT protocol.

### 3.2.3. Blockchain

In our proposed system, a blockchain serves both as a database and an access control system. We use a private Ethereum blockchain with the Clique consensus algorithm. Clique is a proof of authority algorithm and does not require the computational work of the typical Ethereum proof of work algorithm (before the Ethereum merge).

The access control functionality of our system is implemented via smart contracts on the blockchain. There are three smart contracts in our system: Registration Contract (RC), Visitor Contract (VC), and Access Policies Contract (APC). The RC and the APC are deployed to the blockchain by the authorized officials in the system The RC's only purpose is to register new visitors and therefore only has one main function, Register. The Register function takes three arguments, namely, the access level of the visitor, the first published hash of a visitor's hashchain, and an identifier for the visitor, and creates a visitor contract on the chain. The Register function will do so only if the request comes from an authorized account, the management account in our case. If the transaction which accessed the Register function was not sent from a management account, then the request is ignored, and a security alert is issued. Figure 5 shows the steps of the registration process and Figure 6 shows the code for the Register function of the Registration Contract.

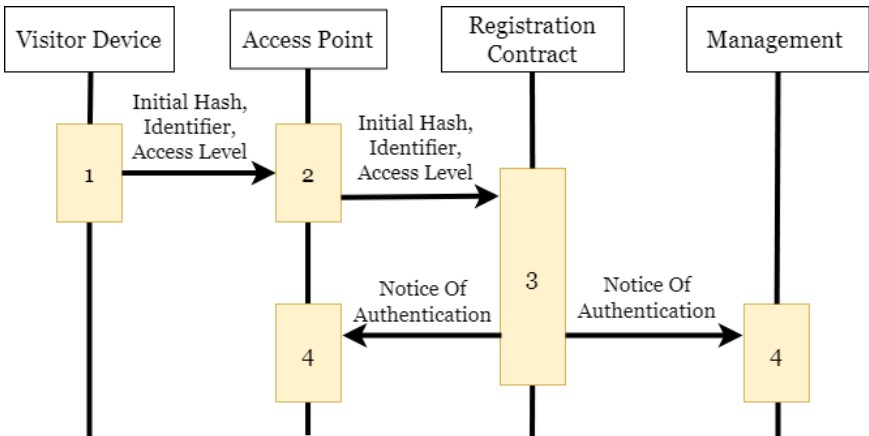

**Figure 5.** Registration process for our authentication system.

```
1  modifier onlyManager(){
2      require(manager == msg.sender);
3      _;
4  }
5
6  function Register(bytes initHash, uint accessLevel, string identifier) public onlyManager(){
7
8      Visitor newVisitor = New Visitor(initHash, accessLevel, identifier, regLat, regLong);
9      emit event VisitorCreated(address(newVisitor), initHash);
10
11 }
```

**Figure 6.** Listing for the Register function of Registration Contract.

An instance of a VC stores all the data for one specific visitor. It stores the visitor's access level which determines the locations the visitor is allowed to enter. It stores the current and previous locations of its paired visitor device and provides a method called updateLocation (provided in Figure 4) to submit new locations. The updateLocation function is the only way to update the current location stored in a visitor contract. As it was mentioned in the Visitor Device description, the VC also stores a published hash value that is used to authenticate that incoming messages are from the corresponding visitor device. Finally, the VC also has a method called RequestNewLocation that emits an event to request the paired visitor device send a location update. The communication for the request is done via an access point. Figure 7 shows this process.

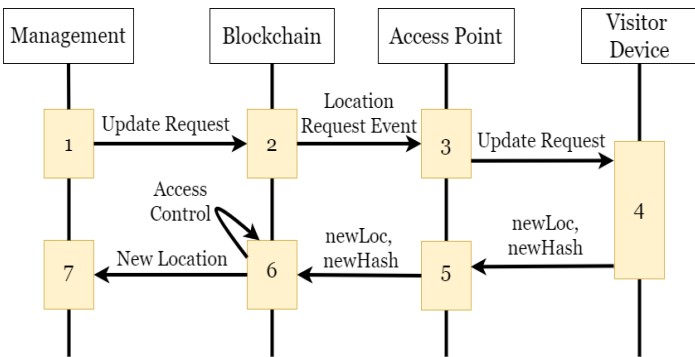

**Figure 7.** Diagram describing a location update request to a visitor device.

The final contract, APC, maintains access levels for locations. It has a simple method CheckAccess which takes a latitude-longitude pair and a visitor's access level as input. Latitudes and longitudes are put into a mapping function and the access level for that area is found. If the visitor access level provided is not valid for the location, then an alert is sent to the Management system. If the access level is correct, then the visitor contract emits an event confirming its new location. It is important to note that when a VC updates its location, it automatically submits to the APC to validate its location. Since checking of access is done on the blockchain, an attacker is unable to stop the verifications or modify the transactions sent to the APC. Figure 8 shows the implementation of the CheckAccess function in our system.

```
1   function CheckAccess(float lat, float long, int accessLevel){
2       int locationID = findID(lat, long);
3       //findID() is simply some mapping from latitude and longitude to a location id
4
5       if(LocationPolicy[locationID] <= accessLevel){
6           emit securityAlert();
7       }
8       else{
9           return true;
10      }
11  }
```

**Figure 8.** Listing of CheckAccess method from APC.

### 3.2.4. Management System

The management system communicates with the blockchain primarily through events emitted from smart contracts. When the registration contract registers a new user, it emits an event containing the visitor's information which the management system then records. After a visitor contract receives a location update an event is emitted which the management system logs. If too much time has passed since a visitor has updated their location, the management system calls a method in the visitor's contract which will prompt

the corresponding visitor device to update its recorded location. After such a request has been sent, management waits for the visitor contract to emit an event giving notice of a new location. If a window of time passes and the management system does not detect this event, security is alerted. Note that the visitor device does not control the interval at which its location is sent to the blockchain. Security is also alerted if the access policies contract emits an event that a visitor is in an unauthorized location. We implemented this process in a JavaScript file. The frequency of requests and the timings for waiting for a response can be specified according to the use case's security requirements.

## 4. Results

We conducted experiments on a private Ethereum blockchain with two full nodes. One node ran on a laptop with an Intel i7-11375H microprocessor at 3.30 GHz clock rate, and the other on a Raspberry Pi 400 computer with a Broadcom BCM2711 quad-core Cortex-A72 microprocessor running at 1.8 GHz clock rate. The Raspberry Pi also served as an access point, running a Python flask server in addition to its blockchain node. The laptop served as the management system, running a JavaScript file in addition to the blockchain node. Simulated visitors were also run on the laptop with JavaScript code. Figure 9 shows our experimental setup. We collected data on three aspects of our system: 1. The time to register a new visitor; 2. The time to update a visitor's location; and 3. The storage required of the system. In this section we present our results.

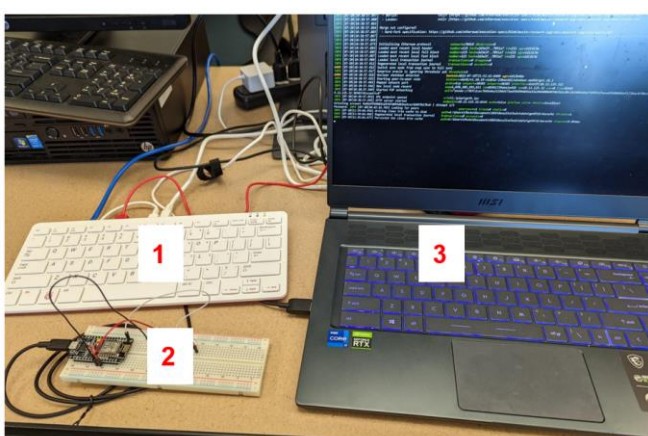

**Figure 9.** Experimental setup, 1. Access Point, 2. Visitor Device, 3. Management Server.

Registration Time: Registration time is measured as the time it takes from the start of a visitor device creating its hash chain to the detection of a new visitor contract by the management system. We tested this with three different experiments. The first experiment measured the time for one physical device to register. The NodeMCU prototype initiated the registration process, and we measured the time between the NodeMCU first requesting registration and when the corresponding block on the blockchain was mined. We repeated this process 10 times and the average of those trials is presented in Table 1. The second experiment repeated the process as the previous, except, instead of a NodeMC, a singular simulated visitor device was used. Finally, in the third experiment, 50 simulated visitors simultaneously requested the registration of visitors. In this experiment, we measured the time between when the first request was sent to the blockchain and the block containing the contract for the 50th visitor was mined. When just comparing one physical device to one simulated device, the physical device was 25% slower to register, though both had an average registration time of under 150 milliseconds. For 50 simulated visitors, the registration time per device is lower due to the fact that the very first visitor registration will take up some time for startup that the later registrations will not need. This results in lower registration times and hence lower average registration times.

**Table 1.** Average registration time per visitor, rounded to the nearest millisecond.

| 1 Physical Visitor Device | 1 Simulated Visitor Device | 50 Simulated Visitors |
|:---:|:---:|:---:|
| 140 | 112 | 87 |

Location Update Time: Location update time was measured as the time elapsed from when a request by the management system to a visitor contract is sent, to when the management system detects a successful update event from the blockchain. The results of our measurements are summarized in Table 2. The first two columns show the average for all measurements collected in a 10 min interval for one device, one physical prototype, and one simulated. The third column shows the average of the collected measurements for one physical device when there were 50 simulated visitors also on the network. The data shows that the response time for more frequent requests is higher in each case. In the case of one physical NodeMCU, the increase in location update time is 9%. The increase is about 19% for one simulated visitor device and 5% with 50 simulated devices. This indicates that the frequency of the requests impacts the location update times so needs to be taken into account when setting up the system.

**Table 2.** Average time to update a location in milliseconds. Data were collected during 10 min trials, with updates at 15 s intervals and 60 s intervals.

| Frequency of Requests | 1 Physical Visitor Device | 1 Simulated Visitor Device | 1 Physical Device W/50 Simulated Visitors Network Load |
|:---:|:---:|:---:|:---:|
| Requests every 15 s | 1020 | 133 | 1050 |
| Requests every 60 s | 936 | 166 | 1002 |

Figures 10 and 11 show the behavior of the system with respect to location updates. The figures show the recorded response times of our physical visitor device over a 10 min testing interval in two situations: (a) when it was the only device on the network; and (b) when there were 50 simulated visitors on the network with it. The plots in Figure 10 show three trials in each of these scenarios when requests were sent at 15 s intervals. Figure 11 is similar, though requests were sent at 60 s intervals. The plots show that the first location update requires some extra time. After the first update, the time needed for location updates falls within a range of 500–1500 milliseconds timeframe for most measurements. Over all the data collected, we saw no response over 4000 milliseconds. Therefore, in our experiments the management system was configured to raise an alarm if 5 s elapsed since sending a request without a response. This threshold value can be configured in the system according to the requirements and needs of the application.

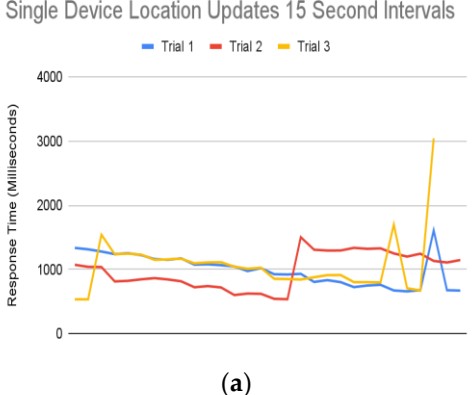
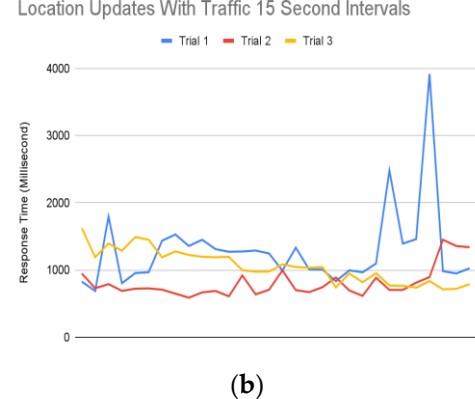

(**a**)                      (**b**)

**Figure 10.** Time to update visitor location at 15 s request intervals over a period of 10 min: (**a**) Response times for a physical device alone on the network over three trials; (**b**) Response times for a physical device with 50 simulated virtual visitors on the network.

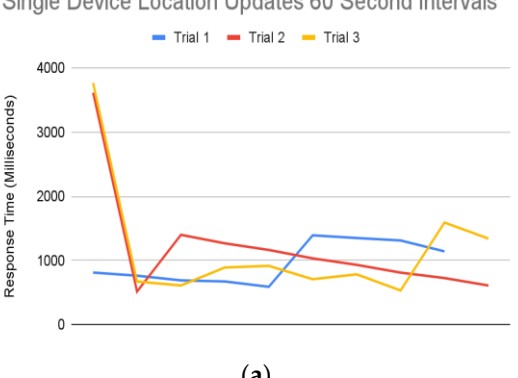

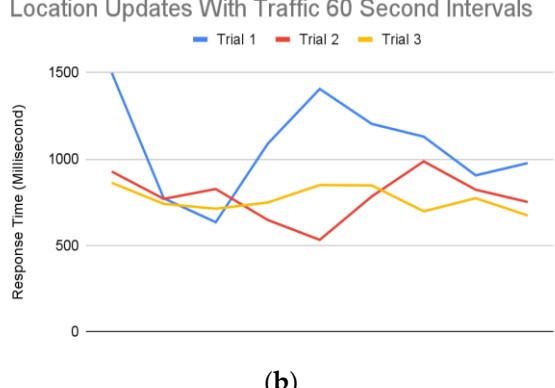

(**a**)  (**b**)

**Figure 11.** Time to update visitor location at 60 s request intervals over a period of 10 min: (**a**) Response times for a physical device alone on the network over three trials; (**b**) Response times for a physical device with 50 simulated virtual visitors on the network.

In both Figures 10 and 11, a comparison of the update location times in scenarios (a) and (b) tells us that the location update time is not significantly impacted by how many visitors are in the system. Even when there were 50 additional visitors in the system, the measured response times stayed stable on the most part. This shows that our system is scalable, and that blockchain-based systems for real-world physical access control applications are feasible. Another encouraging observation is that the location updates time measurements are not significantly impacted by the frequency of the requests from each visitor. There is a threshold that needs to be respected when setting up the system as discussed previously, but the increase from one request per visitor per minute to four requests per visitor per minute does not significantly impact the location update time. This also hints at the scalability of the system for practical applications in which checking access frequently is crucial.

Data Storage: Measuring how much data is used by our system was accomplished by recording the size of the "chaindata" subdirectory found inside of the data directory on our blockchain nodes. The size of the directory was recorded at the start of the experiment and then again at the end of the experiment, the difference between the two measurements was recorded as the storage usage for that scenario. We recorded storage usage of two scenarios, the first being the registration of a new user, and the second was the registration of new users followed by 10 min of location updates at two different intervals (60 and 15 s). The data from these experiments are summarized in Tables 3 and 4, respectively. These results were consistent across the different experiments; thus the type of device and number of devices do not affect the data stored per visitor. The size of data is determined by the implementation of the relevant smart contracts. We note that the storage required per visitor is significantly less when there were 50 visitors on the network as compared to only one. This seems paradoxical as there are more transactions, however, when there is only one visitor, each block added to the chain contains only one transaction, the new location. When multiple visitors submit transactions to the blockchain at roughly the same time, they can be bundled into the same block, making the data usage more efficient as fewer blocks can be used, minimizing the ratio of block metadata to new locations.

**Table 3.** Average storage used to register visitors, rounded to the nearest byte.

|  | 1 Physical Visitor Device | 1 Simulated Visitor Device | 50 Simulated Visitors |
|---|---|---|---|
| Average Storage Used | 5520 | 5500 | 5420 |

**Table 4.** Storage usage in Bytes to store 10 min of location updates.

| Frequency of Requests | 1 Physical Visitor Device | 1 Simulated Visitor Device | 1 Physical Device W/ 50 Simulated Visitors Network Load |
|---|---|---|---|
| Requests every 15 s | 93,908 | 93,834 | 4,992,656 |
| Requests every 60 s | 32,993 | 32,993 | 1,667,138 |

## 5. Security Analysis

In this section we consider ways an adversary may attempt to compromise the system and the mechanisms we have in place to prevent these attacks.

### 5.1. Creating Unauthorized Visitor Contracts

In the proposed system registration of visitors is done via the registration contract that lives on the blockchain. A visitor contract that is not created by the registration contract will not be registered by the management system or the access points. The registration contract will not honor any requests for registration unless they come from an authorized management account, therefore, an attacker must break the cryptographic functions used to generate management accounts if they wish to create unauthorized visitors.

### 5.2. Impersonating a Visitor

In the proposed system, a visitor transaction is invalid if it is not accompanied by a value that hashes to the visitor's most recently published element of its hash chain. So, impersonating a visitor would require knowledge of the hash values in the chain. These values are stored locally, and, unless intercepted during transmission, are not stored elsewhere. To predict the identifying value from the published hash value would amount to 'cracking' the underlying hash algorithm. Since the underlying hash algorithm is cryptographically strong, this would be infeasible.

Another impersonation attack could involve the attacker registering their own "device" on the blockchain. In the current implementation, this is possible because, even though the registration contract is accessible only by management accounts, there is currently no verification of the device sending the registration request between the access point and the visitor device. This can be addressed in various ways. A proposed solution would be to delegate the creation of visitor hash chains to the management system. Before a visitor device is given to a visitor, the management system creates a visitor contract, using the final element of the chain, and securely transmits the hash chain to the visitor device along with the address of this new contract, so that the device may access its corresponding contract. With this system, an attacker would have to gain access to a management account to register their own visitor device. This would require breaking the encryption algorithms utilized by the blockchain, which is assumed to be computationally infeasible. Another solution would be to include a unique identifier with each visitor device and have the active identifiers stored on the registration contract. When a registration request comes from a device with an inactive or illegitimate id, it will not be honored by the registration contract.

### 5.3. Tampering with Records

An attacker may attempt to tamper with transactions on the network to change the records of previous visitor locations or change the access level for the visitor device to a higher access level than what it should be. In the proposed system the location data as well as the access levels are stored on the blockchain and updates to data can only be done by authorized accounts. The location updates are sent by visitor devices and all communication from the visitor device is authenticated using the hash chain on the visitor device. Tampering with the hash chain would result in no updates being submitted to the blockchain, and in turn, to the management system, and hence an alert would be issued about the incident.

Physical tampering of the visitor device would require the breaking of the bracelet which would notify the blockchain via the access point. In such a case, an event would be emitted by the visitor contract notifying the management of the incident.

### 5.4. Unauthorized Access between Updates

In the proposed system visitors update their locations at set intervals. Theoretically, it would be possible for a visitor to walk into an area they are not allowed in right after an update and leave before their device updates their location on the blockchain. The time interval between updates can be set by the management based on the topography of the area in which this system is implemented. Location updates can be set to be relatively frequent making it practically impossible for a visitor to linger in any unauthorized area long enough to do harm without their location being reported.

### 5.5. Replay Attacks

In a replay attack the attacker intercepts the communication between two devices and fraudulently delays or resends it to the receiver. In our scenario the communication between the visitor device and the access point is susceptible to replay attacks. The rest of the communication in the system goes on the blockchain and hence replay attacks are not feasible.

In our proposed system, an attacker without a visitor device may conduct a replay attack to present itself as a legitimate visitor to the access point. This requires that the attacker authenticates itself to the access point. Because the authentication requires a previously calculated hash value to be transmitted with each message, the attacker could intercept the communication including the hash values and replays the hash values for authentication. One solution to this may be an implementation of a timeout system like the one proposed by Li [28] where transactions are forced to timeout after a carefully selected, short time frame. To successfully launch a replay attack, the access point would send twice the number of messages if an attacker is relaying and replying to a visitor. Thus, creating a timeout after a short interval would diminish a replay attack because timeouts would happen before the attacker has time to complete the computations required to replay a visitor: an attacker would need to finish the double amount of computations before the timeout. The timeout would hinder attackers without harming legitimate transactions. However, the timeout would need to be set up correctly for this countermeasure to work. This is left as a future work.

## 6. Discussion

Our study shows that blockchain based access control for a physical site is feasible and has advantages over traditional decentralized systems, including the removal of the control from a decentralized system protects the system against a single point of failure, the use of the cryptographically chained blocks to avoid tampering from outsider and insider attacks, the tamper-proof ledger that records all transactions to provide reliable logging and auditing for the system, and finally, the automated registration of visitors and location updates using smart contracts to reduce the authorization and management tasks, while at the same time providing flexibility for more fine-grained access control if needed.

Additionally, the results from our experiments indicate a flexible and scalable system where the number of visitors and the frequency of location updates do not significantly impact the efficiency of the system. A challenge for real-life applications might be the limited storage capacity of access points, which depends on the number of visitors and the frequency of location updates. One solution to this challenge is to allow for light nodes (instead of full-nodes) on the blockchain. Light nodes can store a certain number of recent blocks instead of all blocks thereby reducing the storage requirement. This solution is viable for large geographical sites with many access points, some of which can serve the full nodes and some others as light nodes.

Although our current work models one physical site under a single authority with several nodes, the advantages of using a blockchain for visitor authentication and access control can be amplified when a consortium of sites (with multiple managing authorities) wants to track users' locations and behaviors. The decentralized nature of the blockchain allows consortium members to reliably track visitors without having to trust each other's practices.

## 7. Conclusions and Future Work

In this study, we propose a system that deploys blockchain technology for physical visitor access control and authentication. We also constructed a visitor device that is easy and relatively cheap to build to provide user authentication and tracking. Our results show that a blockchain-based physical access control system is feasible and scalable.

Future work can include improvements on the design of the wearable bracelet, study on accessibility issues, and improvements and considerations when deploying the physical authentication over a consortium (instead of a single authority) with light/full blockchain nodes.

**Author Contributions:** Conceptualization, Y.K.P.; methodology, Y.K.P.; software, F.S., J.H. and Y.K.P.; validation, F.S., J.H. and Y.K.P.; resources, Y.K.P. and A.J.P.; writing—original draft preparation, F.S., J.H. and Y.K.P.; writing—review and editing, A.J.P. and Y.K.P.; supervision, Y.K.P.; project administration, Y.K.P. and A.J.P.; funding acquisition, A.J.P. and Y.K.P. All authors have read and agreed to the published version of the manuscript.

**Funding:** This research was supported by the U.S. National Science Foundation under grant award # 1950416.

**Institutional Review Board Statement:** Not applicable.

**Informed Consent Statement:** Not applicable.

**Data Availability Statement:** Not applicable.

**Acknowledgments:** We thank the anonymous reviewers who have helped us to improve the content, organization, and presentation of this work.

**Conflicts of Interest:** The authors declare no conflict of interest.

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
