# Peer review of "Physical Visitor Access Control and Authentication Using Blockchain, Smart Contracts and Internet of Things"

_cryptography, doi:10.3390/cryptography6040065_

Round 1

Reviewer 1 Report

The paper proposes a system that deploys blockchain technology to provide visitor authentication and access control in a physical environment A visitor device (i.e. a bracelet) has been constructed for users' authentication.

Note that the concepts behind blockchain are well-known by the research community, so authors do not need to describe blockchain functionalities in the introduction.

The state of the art is too limited (only 13 citations) and should be revised (some references lack the year, for example). It is worth to remark that the related works help in understanding the paper's contribution. In the conclusion, the authors state that studies on blockchain-based access control systems for physical locations are rare: maybe it is better to specify this aspect early in the paper.

Figure 1 should be revised since it is not so clear how the system is structured and how it scales. Moreover, please provide more details about the implementation and structure of the smart contracts.

Besides tables, it would be useful to provide a graphical representation of the results, also to show the behavior of the system during the time. Furthermore, it would be interesting to give information about the duration of the experiments.

Author Response

We'd like to thank the reviewer for their valuable comments. Our responses to their comments are typed in blue. Our revised paper is  included in the attached document.

Note that the concepts behind blockchain are well-known by the research community, so authors do not need to describe blockchain functionalities in the introduction.

The introduction is updated to eliminate the descriptions of blockchain functionalities. 

The state of the art is too limited (only 13 citations) and should be revised (some references lack the year, for example). It is worth to remark that the related works help in understanding the paper's contribution.

The state of the art (the  Related Works section)  is expanded to include more of the relevant literature and the missing fields in the refrences are added.

In the conclusion, the authors state that studies on blockchain-based access control systems for physical locations are rare: maybe it is better to specify this aspect early in the paper.

The main contributions of the study are clearly stated at the end of the revised Introduction section.

Figure 1 should be revised since it is not so clear how the system is structured and how it scales.

An additional diagram showing the overview of the  system is added and how the various components of the system  work together is elaborated.

 Moreover, please provide more details about the implementation and structure of the smart contracts.

Details about the implementation are added in section 3.2 Implementation. Snippets of code for some of the essential functions are provided when deemed necessary.

Besides tables, it would be useful to provide a graphical representation of the results, also to show the behavior of the system during the time. Furthermore, it would be interesting to give information about the duration of the experiments.

Graphs depicting the behavior of the system over time are included. The duration of the experiments are made more clear in the revised Results section.

Reviewer 2 Report

This research paper presents the use of a “bracelet” based on a low-cost NodeMCU Internet of Things (IoT) platform that broadcasts visitor location information and cannot be removed without alerting a management system. 

Here I list several concerns.

1. The organization of the introduction is not well.  you should conclude your innovation and contribution at the end of the introduction.

2. The "Discussion" section is missing, it should be added.

3. Discuss applicability and implementation in real-life scenarios.

4. How does authentication work?

5. Figure 1 requires a detailed explanation, such as how blockchain is integrated with current research?

6. Some citations are very old. Authors could add few updated and relative papers such as https://doi.org/10.1109/JIOT.2020.2969326; DOI: 10.1109/MNET.001.2100412; DOI: 10.1109/TNSE.2022.3186393. Just to name some.

Author Response

We'd like to thank the reviewer for their valuable comments. Our responses to their comments are typed in blue. Our revised paper is  included in the attached document.

  1. The organization of the introduction is not well.  you should conclude your innovation and contribution at the end of the introduction.

The introduction section is updated and our contribution is included at the end of the Introduction.

  1. The "Discussion" section is missing, it should be added.

The Conclusion and Future Works section is expanded to include a discussion of the results and renamed as  “Discussion”.

  1. Discuss applicability and implementation in real-life scenarios.

The new Discussion includes a discussion of the applicability of the proposed system in real-life applications.

  1. How does authentication work?

The authentication process is elaborated in section 3.1 Proposed System  and 3.2.1 Visitor Devices.

 Figure 1 requires a detailed explanation, such as how blockchain is integrated with current research?

An additional diagram showing the overview of the  system is added and how the various components, including the blockchain, work together is elaborated.

  1. Some citations are very old. Authors could add few updated and relative papers such as https://doi.org/10.1109/JIOT.2020.2969326; DOI: 10.1109/MNET.001.2100412; DOI: 10.1109/TNSE.2022.3186393. Just to name some.

The Related Works section is expanded to include more of the relevant literature. The relevant ones among the articles suggested  by the reviewer are also included. In particular, https://doi.org/10.1109/JIOT.2020.2969326  (A Survey on Access Control in the Age of Internet of Things) is cited. 

10.1109/TNSE.2022.3186393 (Privacy Protection Method Based on Multidimensional Feature Fusion Under 6G Networks)  and  10.1109/MNET.001.2100412 (Large-Capacity Local Multi-Dimensional Information Hiding Method for 6G Networks) are not cited as they are not closely related to the topic of the study.

Reviewer 3 Report

This study develops a visitor authentication and access control in a physical environment by using blockchain with IoT devices, and shows the feasibility of their implementation and performance improvements. It is a novel and interesting research issue. However, some descriptions need to be improved.

1. The proposed visitor authentication and access control in a physical environment is developed by using blockchain with IoT devices. Its advantages and limitations should be clearly described.

2. The proposed mechanism should compare the security properties and efficiency with the traditional access control and authentication mechanisms.

3. The main contributions of this study should be clearly listed and highlighted.

Author Response

We'd like to thank the reviewer for their valuable comments. Our responses to their comments are typed in blue. Our revised paper is  included in the attached document.

  1. The proposed visitor authentication and access control in a physical environment is developed by using blockchain with IoT devices. Its advantages and limitations should be clearly described.

The advantages of the system are elaborated in the introduction as well as in the new Discussion section.

  1. The proposed mechanism should compare the security properties and efficiency with the traditional access control and authentication mechanisms.

A comparison of security properties and efficiency with the traditional access control and authentication mechanisms is included in the new Discussion section

  1. The main contributions of this study should be clearly listed and highlighted.

The main contributions of the study are clearly stated at the end of the revised Introduction section.

Round 2

Reviewer 1 Report

The authors carefully addressed the required changes.

Author Response

Thank you for your valuable comments to improve our work.

Reviewer 2 Report

Better explanations are needed, such as how does blockchain integrate with current research?

Author Response

Dear reviewer and editors,

We carefully reviewed your comments from the first round of reviews, and we believe we addressed them. However, when we reviewed your second round of reviews to make improvements, we saw that the scores seem to be lower than in the first round of reviews, which comes as a surprise to us, as we believe we addressed them (see answers to your first round of review below).

Also this comment in the second round of reviews:

Better explanations are needed, such as how does blockchain integrate with current research?

This comment we believe does not help us to improve the content of the paper, as "current research" is not specific to the content of the paper. Blockchain is a current research topic on its own, and applications of this technology are widespread in many other current research areas in information technology, business, and others. The introduction of our paper mention some of the current applications of blockchain.

We also believe that your comments in the first round of reviews were similar to the others submitted by the two other reviewers that we carefully addressed in the submission and that they stated that we indeed addressed them to their satisfaction.

Thank you for your time and comments.

Best regards,

The authors.

-------------------Start reviews-----------------------

We'd like to thank the reviewer for their valuable comments. Our responses to their comments are typed in blue. Our revised paper is included in the attached document.

  1. The organization of the introduction is not well.  you should conclude your innovation and contribution at the end of the introduction.

The introduction section is updated and our contribution is included at the end of the Introduction.

  1. The "Discussion" section is missing, it should be added.

The Conclusion and Future Works section is expanded to include a discussion of the results and renamed as  “Discussion”.

  1. Discuss applicability and implementation in real-life scenarios.

The new Discussion includes a discussion of the applicability of the proposed system in real-life applications.

  1. How does authentication work?

The authentication process is elaborated in section 3.1 Proposed System  and 3.2.1 Visitor Devices.

 Figure 1 requires a detailed explanation, such as how blockchain is integrated with current research?

An additional diagram showing the overview of the  system is added and how the various components, including the blockchain, work together is elaborated.

  1. Some citations are very old. Authors could add few updated and relative papers such as https://doi.org/10.1109/JIOT.2020.2969326; DOI: 10.1109/MNET.001.2100412; DOI: 10.1109/TNSE.2022.3186393. Just to name some.

The Related Works section is expanded to include more of the relevant literature. The relevant ones among the articles suggested by the reviewer are also included. In particular, https://doi.org/10.1109/JIOT.2020.2969326  (A Survey on Access Control in the Age of Internet of Things) is cited. 

10.1109/TNSE.2022.3186393 (Privacy Protection Method Based on Multidimensional Feature Fusion Under 6G Networks)  and  10.1109/MNET.001.2100412 (Large-Capacity Local Multi-Dimensional Information Hiding Method for 6G Networks) are not cited as they are not closely related to the topic of the study.

Reviewer 3 Report

The authors revised the manuscript and addressed previous issues in this version. There are no further comments for this article.

Author Response

(The authors gave the same response as above.)
